# Edges are all you need: Potential of medical time series analysis on complete blood count data with graph neural networks

Daniel Walke[1,2*], Daniel Steinbach[3,4], Sebastian Gibb[3,5], Thorsten Kaiser[6], Gunter Saake[2], Paul C. Ahrens[3], David Broneske[7‡], Robert Heyer[8,9‡]

**1** Bioprocess Engineering, Otto von Guericke University, Universitätsplatz 2, Magdeburg, Germany, **2** Database and Software Engineering Group, Otto von Guericke University, Universitätsplatz 2, Magdeburg, Germany, **3** Institute of Laboratory Medicine, Clinical Chemistry and Molecular Diagnostics, Leipzig University Hospital, Leipzig, Germany, **4** Medical Informatics Center—Department for Clinical AI and Translational Medicine, University of Leipzig Medical Center, Leipzig, Germany, **5** Anesthesiology and Intensive Care Medicine, University Medicine Greifswald, Greifswald, Germany, **6** University Institute for Laboratory Medicine, Microbiology and Clinical Pathobiochemistry, OWL University Hospital of Bielefeld University, Detmold, Germany, **7** German Centre for Higher Education Research and Science Studies (DZHW), Lange Laube 12, Hannover, Germany, **8** Multidimensional Omics Analyses group, Leibniz-Institut für Analytische Wissenschaften – ISAS – e.V., Bunsen-Kirchhoff-Straße 11, Dortmund, Germany, **9** Faculty of Technology, Bielefeld University, Universitätsstraße 25, Bielefeld, Germany

‡ Shared last authors.
* daniel.walke@ovgu.de

## Abstract

### Purpose

Machine learning is a powerful tool to develop algorithms for clinical diagnosis. However, standard machine learning algorithms are not perfectly suited for clinical data since the data are interconnected and may contain time series. As shown for recommender systems and molecular property predictions, Graph Neural Networks (GNNs) may represent a powerful alternative to exploit the inherently graph-based properties of clinical data. The main goal of this study is to evaluate when GNNs represent a valuable alternative for analyzing large clinical data from the clinical routine on the example of Complete Blood Count Data.

### Methods

In this study, we evaluated the performance and time consumption of several GNNs (e.g., Graph Attention Networks) on similarity graphs compared to simpler, state-of-the-art machine learning algorithms (e.g., XGBoost) on the classification of sepsis from blood count data as well as the importance and slope of each feature for the final classification. Additionally, we connected complete blood count samples of the same patient based on their measured time (patient-centric graphs) to incorporate time series information in the GNNs. As our main evaluation metric, we used the Area Under Receiver Operating Curve (AUROC) to have a threshold independent metric that can handle class imbalance.

**Data availability statement:** All Jupyter Notebooks, python files and datasets of the methodology developed and used in this study are available at https://github.com/danielwalke/SBCDataAnalysis. The used dataset containing the complete blood counts is available in zenodo under https://zenodo.org/records/6922968 (DOI: 10.5281/zenodo.10122491).

**Funding:** Grant-Numbers: HE 8077/2-1, SA 465/53-1 Funder: German Research Foundation/ Deutsche Forschungsgemeinschaft (DFG) (https://www.dfg.de/de). Funded Authors: D. W. The funders had no role in study design, data collection and analysis, decision to publish, or preparation of the manuscript.

**Competing interests:** The authors declare that they have no competing interests.

## Results and Conclusion

Standard GNNs on evaluated similarity-graphs achieved an Area Under Receiver Operating Curve (AUROC) of up to 0.8747 comparable to the performance of ensemble-based machine learning algorithms and a neural network. However, our integration of time series information using patient-centric graphs with GNNs achieved a superior AUROC of up to 0.9565. Finally, we discovered that feature slope and importance highly differ between trained algorithms (e.g., XGBoost and GNN) on the same data basis.

---

## 1. Introduction

Recently, artificial intelligence (AI) showed its great potential in several biological and medical applications, such as diagnosing heart diseases [1] and chronic kidney disease from input matrices with clinical data [2]. For such classification and prediction tasks, researchers proposed several modern state-of-the-art machine learning algorithms (e.g., XGBoost [3]) within the last decades. However, real-world data such as medical data are often connected (e.g., time-dependent measurements of the same patient) [4]. These connections can carry valuable information which helps in increasing the predictive power of machine learning algorithms. However, this information is mostly neglected by state-of-the-art machine learning algorithms [3,5] since they consider data points as independent. Graphs are data structures that can store such connections. A graph G is a non-empty finite set of elements called nodes V(G) and finite set E(G) of distinct unordered pairs of distinct elements of V(G) called edges [6]. Each node and each edge can have features (attributes) attached for a more detailed data characterization. Furthermore, graphs can either have one node type and edge type (homogeneous graph) or multiple node and edge types (heterogeneous graph). An example for a homogeneous graph is a medical graph containing patients with attached features (e.g., age and lab measurements) as nodes connected by edges based on their similarity. In a heterogeneous graph, we store additional features (e.g., lab features) as separate node types and connect patient nodes with their respective feature nodes.

Graphs can be analyzed using graph learning with several algorithms, such as, Graph Neural Networks (GNNs) [7–12]. GNNs sample information (i.e., features) from neighboring nodes, transform this information (e.g., linear transformation with a subsequent activation function), and finally aggregate (e.g., averaging) the transformed information. Sampling, transformation, and aggregation are performed for each node and repeated for a predefined number of iterations (i.e., GNN layers). While all GNNs are based on these steps (i.e., sampling, transformation, and aggregation), they can differ in their architecture (e.g., different sampling and aggregation strategies or by the use of attention mechanisms [13]) [14]. GNNs have the advantage that they can utilize attached features and parallelize computations on modern hardware (e.g., GPUs) in contrast to using embedding techniques like DeepWalk [15] or Node2Vec [16]. They already showed promising results for predicting pediatric

sepsis based on several groups of laboratory tests (e.g., medical history and serological tests) using similarity graphs [17]. However, there are currently two main challenges for the application of GNNs on medical data:

A. Although GNNs showed great potential in diverse applications [18–20] including medical applications [21–24], the performance like Area under Receiver Operating Curve (AUROC) of GNNs in applications for clinical decision support solely based on complete blood count (i.e., hemoglobin, red blood cells, white blood cells, platelets and mean corpuscular volume) data in adults from the clinical routine is currently unclear. Furthermore, GNNs might facilitate and improve the analysis of time series information compared to current deep learning approaches like LSTMs, 1D-CNNs and Transformer-decoder models by natively (i.e., without additional padding and masking of input data) supporting time series of varying length.

B. Although interpretability mechanisms exist to estimate the importance of individual features for machine learning algorithms and GNNs, to the best of our knowledge there is no framework revealing the influence of individual features' directions to the predictions. However, such partial dependencies (e.g., increased sepsis risk for high white blood cell count) are even more important and valuable for clinical applications.

To overcome the unclear performance on complete blood count data from the clinical routine (challenge A), we evaluated the performance of GNNs (Graph SAGE [7], Graph Attention Networks [8], Graph Attention Networks version 2 [25], Graph Isomorphism Networks [10], Heterogeneous Graph Transformer [12], and Heterogeneous Attention Networks [11]) on medical data [26,27] against shallow and ensemble-based machine learning algorithms and a neural network. The selected dataset [26] contains complete blood count data (five blood parameters and additional age and sex) classified as "sepsis" or "control" ("not sepsis"). Sepsis is a life-threatening organ dysfunction caused by a dysregulated immune response to an infection [28]. The inflammatory response is driven through the release of cytokines from neutrophil granulocytes and macrophages. Blood parameters like white blood cells, red blood cells, platelets, hemoglobin and mean corpuscular volume might serve as easily available indicators for sepsis [26] (S1 Note). It is still one of the leading causes of death in critically ill patients worldwide [29,30] and is well-studied [26,31]. An early prediction of sepsis allows fast initiation of an appropriate treatment (e.g., with antibiotics) [32]. GNNs might serve as a useful tool for classifying sepsis based on two different assumptions regarding similarity and incorporation of time-series information:

1. **Similarity:** Instances with similar feature values usually have the same classification labels. Therefore, similarity graphs (a graph connecting instances with similar feature values) might increase the classification performance by potentially connecting sepsis measurements with each other. Applying GNNs on similarity graphs could potentially increase the classification performance compared to other state-of-the-art machine learning algorithms that do make use of similar features.

2. **Time-series information:** Each patient can have multiple measurements at different times during the hospitalization. The time-series information of a single patient could help in identifying sepsis measurements during the stay. These time-series information could be represented as graphs based on measurement times (patient-centric graphs) and analyzed using GNNs.

Furthermore, we evaluated the importance and partial dependence to increase the interpretability of the used models. Therefore, we adopted the partial dependence [33] from scikit-learn [34] to also apply it on GNNs and PyTorch Neural Networks (challenge B).

## 2. Methods

In this section, we describe and explain the workflow of our study to evaluate the performance of graph learning algorithms (Fig 1). First, we pre-processed the complete blood count dataset from Steinbach et al. [26,27] (Fig 1A). Then, we

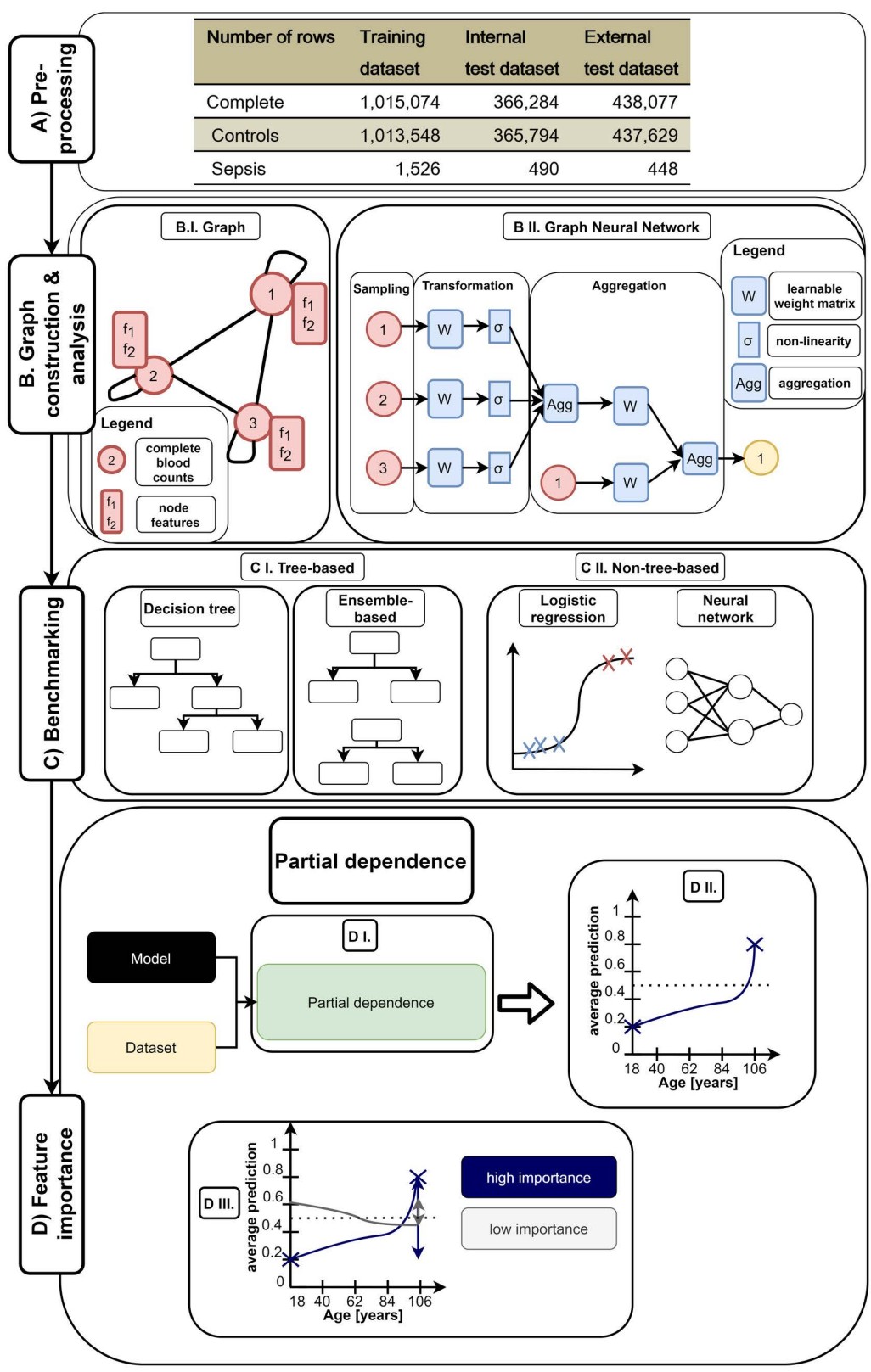

**Fig 1. Workflow of our study to evaluate the performance of GNNs compared to benchmark algorithms.** First the dataset [27] is pre-processed (A) according to the work of Steinbach et al. [26] resulting in a train and two test validation sets (internal and external test set). Then, we constructed

graph structures based on these datasets (B I.) and applied GNNs on them (B II.). GNNs sample information (i.e., features) from neighboring nodes, transform these information (e.g., linear transformation with a subsequent activation function), and finally aggregate (e.g., averaging) the transformed information [14]. Here, we visualized the architecture of one GraphSAGE [7] layer which adds linear transformed features of the target node to the aggregated neighborhood features (B II.). Afterwards, we compare the results of our GNNs against tree-based (Decision Tree, Random Forest, RUS-Boost and XGBoost) (C I.) and non-tree-based (Logistic Regression and a neural network) (C II.) benchmark models. Finally, we evaluated partial dependence of each feature (age, sex, hemoglobin, red blood cells, mean corpuscular volume, white blood cells and platelets) in the dataset to increase interpretability and transparency of the trained models (D). Therefore, we first calculated the average predictions of each feature for various grid values for each model. Then, we plotted resulting average predictions of trained models over all evaluated grid-values for each features. Finally, we calculated and normalized the variance of the features' average predictions among all features for each model to evaluate the influence (i.e., importance) of each feature on the average prediction across all grid values. A high normalized variance (near one) indicates a high importance over different grid values and a low normalized variance (near zero) indicates a low importance.

constructed several graph structures from the dataset and applied GNNs on them (Fig 1B). Afterwards, we benchmarked the GNNs against several other machine learning algorithms and measured their required training time (Fig 1C). Additionally, we evaluated the importance of individual features for the classification of sepsis (Fig 1 D). Finally, we evaluated the performance of Graph Attention Networks (GAT) on several patient-centric graph structures.

## 2.1 Pre-processing and setup

The dataset from Steinbach et al. [26,27] contains patients hospitalized into non-intensive care units from German tertiary care center in Leipzig (internal dataset) and Greifswald (external dataset) between January 2014 and December 2021 [26]. Each patient can have multiple complete blood counts (i.e., rows in the dataset). Each complete blood count measurement contains a patient id, age, biological sex (i.e., only male or female were reported), five blood parameters (hemoglobin, red blood cells, white blood cells, mean corpuscular volume, and platelets), a binary label ("sepsis" or "not sepsis"), and information where and when the measurement was performed. The functions and the potential relevance of the individual blood parameters for sepsis is discussed in the supplement (S1 Note). We pre-processed the dataset and separated it into train, internal and external test sets according to Steinbach et al. [26] (Fig 1A). To visualize the distribution of each feature, we plotted violin plots for each continuous feature (S2 Fig). Afterward, we analyzed the data distribution of each set.

We used the following setup for all analyses:

- Mainboard Supermicro X12SPA-TF

- CPU: Intel® Xeon® Scalable Processor "Ice Lake" Gold 6338, 2.0 GHz, 32 Cores

- GPU: NVIDIA® RTX A6000 (48 GB GDDR6)

- RAM: 8x32 GB DDR4–3200

- ROM: 2TB Samsung SSD 980 PRO, M.2

## 2.2 Graph construction and analysis

After pre-processing, we constructed two similarity graphs from the complete blood count data and analyzed them using GNNs (Fig 1B). The first similarity graph is a homogeneous k-nearest neighbors (k-nn) graph (Fig 4B). It contains a node for each complete blood count measurement and connects them directly based on similarity (normalized Euclidean distance of the features). The second graph is a heterogeneous similarity graph (Fig 4C). It contains a patient sample node for each complete blood measurement and nodes with discretized values (lower and upper limit of the discretization) for each blood parameter as similarity comparison. Discretization was performed based on ten percentiles to have less sensitivity against outliers. Each complete blood count node contains standard normalized patient features (age, sex,

hemoglobin, red blood cells, white blood cells, mean corpuscular volume and platelets) similar to the homogeneous graph. With this heterogeneous graph structure, patients are indirectly connected via similar blood parameters.

The basic assumption behind these graph structures is that similar complete blood counts might have the same label (homophily) and therefore, similarities might increase the classification performance. Afterward, we applied several GNNs (GraphSAGE [7], GAT [8], GATv2 [25], GIN [10], HGT [12], HAN [11]) with two layers (128 neurons) and a learning rate of 0.0003 using PyTorch Geometric [35]. Due to memory constraints while training on the heterogeneous graph, we reduced the size of the hidden dimension of GAT with two attention heads, GATv2 with two attention heads, HGT, and HAN to 64 dimensions. We trained the GNNs for 10,000 epochs with an early stopping after an increase of the validation loss for ten consecutive epochs similar to the work of Kipf and Welling [9]. We chose a high epoch number and relatively low learning rate to guarantee sufficient training (i.e., preventing under-fitting) while also preventing over-fitting by applying early stopping on a separate validation set. All GNNs were evaluated using AUROC, F1- Macro Score and Matthews Correlation Coefficient (MCC).

## 2.3 Benchmarking

As benchmarks, we evaluated the performance (AUROC, F1- Macro Score, and MCC) on tree-based and non-tree-based algorithms (Fig 1C) to get a comprehensive performance evaluation across several algorithms. As tree-based algorithms, we used a Decision Tree and three ensemble-based algorithms (i.e., Random Forest [5], RUSBoost [36], and XGBoost [3]). As non-tree-based algorithms, we used a Logistic Regression and a neural network. The neural network was implemented in PyTorch [37] and used standard normalized complete blood count features with two layers (128 neurons) and a learning rate of 0.0003 similar to the GNNs. The neural network was trained for 10,000 epochs with an early stopping after an increase of the validation loss for ten consecutive epochs. The high epoch number in combination with a low learning rate should prevent under-fitting. Early-stopping on a separate validation set was used to prevent over-fitting. Hyper-parameters for the Logistic Regression and all tree-based algorithms were tuned using grid search with 10-fold cross validation using sklearn (version 1.2.2) [38]. To test the robustness of each benchmark against noise, we added 10 and 100 noisy features (random features between 0 and 1) to the dataset and evaluated the performance of each benchmark algorithm and GraphSAGE [7] as a representative of GNNs.

## 2.4 Partial dependence and feature importance

After performance evaluation, we evaluated the partial dependence [33] of each feature (Fig 1D) for each model to evaluate the influence of different feature values on the average prediction (i.e., the ratio of sepsis prediction). Therefore, we implemented a function similar to scikit-learn [38,39] but with the compatibility for GNNs and required data transformations (e.g., standard normalization). We used a grid resolution of 100 (i.e., 100 different values for each feature) between the 5%-percentile and 95% percentile of each feature. If the number of unique values is below the grid resolution (e.g., sex contains only two discrete values in the input data) only these unique values were used for the evaluation of average prediction. Afterwards, we plotted the average prediction over the respective grid values for each algorithm feature-wise using matplotlib as line charts. For the compatibility of GNNs, we implemented a "predict_proba"-function in each PyTorch-model (neural network and GNNs) which returns the prediction probabilities of each class similar to scikit-learn models. Finally, we calculated the variance of the average prediction over all features to evaluate which features had the highest influence (i.e., importance) among all features for each algorithm. The variance of each feature was normalized over the sum of all variances of the respective algorithm to obtain values between zero (i.e., indicating a low importance) and one (i.e., indicating a high importance). These normalized variance values were the plotted and hierarchically clustered (Euclidean distance) using seaborn's clustermap. We evaluated the performance on all benchmark algorithms and on GraphSAGE (homogenous and heterogeneous) as a representative of GNNs. GraphSAGE was chosen as the final model since it achieved a reliable classification performance on the homogeneous and heterogeneous similarity graphs.

## 2.5 Patient-centric graphs

The constructed similarity graphs are measurement-centric, i.e., they do not consider multiple measurements of the same patient (Fig 4A-C) for the classification. For incorporating multiple measurements of the same patient, we construct several patient-centric graphs (Fig 4D-F). In these graphs, a node represents standard normalized complete blood counts and edges represent connections from previous to following measurements (directed graph, Fig 4D), following to previous measurements (reversed directed graph, Fig 4E) or connecting all measurements of the same patient with each other independent of their order (undirected graph, Fig 4F). Afterward, we added positional encodings on each node to represent the position of the measurement in the sequence of measurements. We applied Graph Attention Networks [8] with two layers (128 neurons), a learning rate of 0.0003, a batch size of 50,000 and trained the GNN for 10,000 epochs with an early stopping after an increase of the validation loss for five consecutive epochs. To prevent under-fitting, we used a high number of epochs and a low learning rate during training. We used early-stopping on a separate validation set again to prevent over-fitting. Then, we compared the classification performance (AUROC) with and without positional encodings for all patient-centric graphs. Finally, we evaluated the attention weights (influence between different nodes) on each graph to increase the interpretability of the trained Graph Attention Networks. Therefore, we returned all attention weights from each layer of the graphs and analyzed mean, standard deviation, and quantiles of nodes connecting to nodes with the same label (e.g., connection of a "control" node to a "control" node) and nodes connecting to nodes with a different label (e.g., connection of a "sepsis" node to a "control" node). Furthermore, we evaluated the classification performance (AUROC) on other GNNs (GraphSAGE [7] and GCN [9]) to evaluate whether the improvement was solely based on the attention mechanism. We trained both GNNs similarly with a learning rate of 0.0003, a batch size of 50,000 and trained the GNN for 10,000 epochs with an early stopping after an increase in the validation loss for five consecutive epochs. Finally, we also compared the performance of our patient-centric GNNs against other deep learning architectures, Long Short-Term Memory (LSTM) [40], bidirectional LSTM (Bi-LSTM) [41], one-dimensional Convolutional Neural Network (1D-CNN) [42] and transformer decoder-only architecture (Transformer) [13]. Since the complete blood count data contains time-series information of different lengths, we padded the time series of each patient to the maximum length and masked paddings from the loss function. We trained each algorithm for 100 epochs with an early stopping after a loss increase over the last two epochs. Hyperparameter tuning (number of layers, hidden dimension, learning rate, weight decay, kernel size on the 1D-CNNs, number of heads on the Transformer) and early stopping was performed using a separate validation dataset.

## 2.6 Ethics statement

The Ethics Committee at the Leipzig University Faculty of Medicine approved the initial study from Steinbach et al. [26] (reference number: 214/18-ek). The study was published in accordance with the Transparent Reporting of a multivariable prediction model for Individual Prognosis or Diagnosis (TRIPOD) statement. This study is only re-evaluating the dataset from Steinbach et al. [26] by evaluating GNNs and incorporating time-series information.

## 3. Results

After data pre-processing (Fig 1A), we evaluated the performance of GNNs on similarity graphs compared to other machine learning algorithms (Fig 1B and C) on two data sets representing the complete blood count for sepsis and non-sepsis patients' data. Afterward, we evaluated the feature slope and importance of different algorithms to increase their transparency and interpretability (Fig 1D). Finally, we created patient-centric graphs and applied the attention mechanisms on these graphs to achieve a superior performance and highlight the importance of an appropriate graph structure for the desired use case.

### 3.1 Performance of graph learning on similarity graphs for classifying complete blood counts

First, we wanted to evaluate the performance of different GNNs on medical data compared to other machine learning algorithms. Therefore, we applied several GNNs (GraphSAGE [7], GAT [8], GATv2 [32], GIN [10], HGT [12], and HAN

[11]) and other machine learning algorithms on sepsis blood count data, evaluated their performance (i.e., AUROC, Matthews Correlation Coefficient, F1-Macro) and their required training time (Table 1). In the following, we will mainly focus on AUROC as the primary evaluation metric to have a threshold-independent evaluation metric that can also incorporate the high class imbalance. By assessing model performance across different thresholds, AUROC enables clinicians to fine-tune the sensitivity and specificity of sepsis detection according to their needs, minimizing both the risk of missing septic patients and the potential harm from overdiagnosis, such as unnecessary antibiotic treatments. Nearly all GNNs revealed a similar performance on the homogeneous similarity graph (AUROC: ≤ 0.8741) and heterogeneous similarity graph (AUROC:

≤0.8747) on both datasets. Furthermore, the neural network (AUROC: ≤ 0.8806), ensemble-based machine learning algorithms (Random Forest (AUROC: ≤ 0.8700), RUSBoost (AUROC: ≤ 0.8680) and XGBoost (AUROC: ≤ 0.8643)) had a similar performance. However, some GNNs (GAT, GATv2, HGT and HAN) on heterogeneous similarity graphs and shallow algorithms, like Logistic Regression (AUROC: ≤ 0.8369) and Cecision Tree (AUROC: ≤ 0.8391), performed much worse on both datasets compared to other GNNs, the neural network and ensemble-based algorithms. Our results of the RUSBoost algorithm (AUROC: ≤ 0.8680) are consistent with the results of Steinbach et al. [26] (AUROC: ≤ 0.872).

**Table 1. Comparison of GNNs against benchmarks on complete blood count data for sepsis classification (higher values represent better performance) and their required training time. We evaluated the classification performance on two datasets (internal and external dataset). Bold values represent the best values in each column. MCC (Matthew's Correlation Coefficient), AUROC (Area under receiver operating curve).**

| Models | | AUROC | | F1-Macro | | MCC | | Training time [s] |
|---|---|---|---|---|---|---|---|---|
| | | Internal | External | Internal | External | Internal | External | |
| **Tree-based benchmarks** | Decision Tree [53] | 0.8391 | 0.7870 | 0.4313 | 0.4018 | 0.0432 | 0.0291 | 2.00 |
| | Random Forest [5] | 0.8700 | **0.8178** | **0.4770** | **0.4609** | **0.0605** | 0.0385 | 17.36 |
| | RUSBoost [36] | 0.8680 | 0.8153 | 0.4701 | 0.4497 | 0.0576 | 0.0361 | 212.88 |
| | XGBoost [3] | 0.8643 | 0.8121 | 0.4373 | 0.4184 | 0.0495 | 0.0327 | **0.54** |
| **Non-tree-based benchmarks** | Logistic regression [54] | 0.8369 | 0.7558 | 0.4412 | 0.3736 | 0.0442 | 0.0222 | 5.97 |
| | Neural Network [55] | **0.8806** | 0.8145 | 0.4479 | 0.4502 | 0.0521 | 0.0383 | 19.97 |
| **Homogeneous graph learning** | Graph SAGE [7] | 0.8741 | 0.8052 | 0.4411 | 0.3964 | 0.0499 | 0.0308 | 394.69 |
| | GAT [8] (single attention head) | 0.8726 | 0.8086 | 0.4440 | 0.4457 | 0.0501 | 0.0374 | 561.23 |
| | GAT [8] (two attention heads) | 0.8707 | 0.8114 | 0.4476 | 0.4502 | 0.0513 | **0.0393** | 1,008.311 |
| | GATv2 [25] (single attention head) | 0.8723 | 0.8057 | 0.4413 | 0.4442 | 0.04889 | 0.0371 | 553.55 |
| | GATv2 [25] (two attention heads) | 0.8746 | 0.8130 | 0.4469 | 0.4500 | 0.0511 | 0.0384 | 1,189.47 |
| | GIN [10] | 0.8649 | 0.8050 | 0.4499 | 0.4530 | 0.0502 | 0.0372 | 31.45 |
| **Heterogeneous graph learning** | Graph SAGE [7] | 0.8747 | 0.8176 | 0.4422 | 0.4020 | 0.0506 | 0.0326 | 981.10 |
| | GAT [8] (single attention head) | 0.8426 | 0.8055 | 0.4396 | 0.4094 | 0.0464 | 0.0328 | 1,086.07 |
| | GAT [8] (two attention heads) * | 0.8396 | 0.8069 | 0.4404 | 0.4164 | 0.0459 | 0.0334 | 679.95 |
| | GATv2 [25] (single attention head) | 0.8402 | 0.8061 | 0.4384 | 0.4136 | 0.0455 | 0.0329 | 323.36 |
| | GATv2 [25] (two attention heads) * | 0.8422 | 0.8067 | 0.4401 | 0.4144 | 0.0468 | 0.0326 | 463.59 |
| | GIN [10] | 0.8696 | 0.8051 | 0.4215 | 0.3626 | 0.0456 | 0.0273 | 54.95 |
| | HGT [12]* | 0.8317 | 0.7778 | 0.4509 | 0.4197 | 0.0479 | 0.0289 | 2,696.46 |
| | HAN [11]* | 0.8401 | 0.8036 | 0.4397 | 0.4106 | 0.0457 | 0.0328 | 13,039.06 |
| **Minimum** | | 0.8317 | 0.7558 | 0.4215 | 0.3626 | 0.0432 | 0.0222 | 0.54 |
| **Mean** | | 0.8584 | 0.8040 | 0.4444 | 0.4221 | 0.0491 | 0.0335 | 1,166.10 |
| **Maximum** | | 0.8806 | 0.8178 | 0.4770 | 0.4609 | 0.0605 | 0.0393 | 13,039.06 |

*Hidden dimension reduced to 64 due to memory constraints.

XGBoost had the lowest required training time (0.54 s) followed by the Decision Tree (2.00 s) and Logistic Regression (5.97 s). The Random Forest (17.36 s) and one GNN (GIN: ≤ 54.95 s) are faster than the RUSBoost ensemble algorithm (212.88 s). However, nearly all homogeneous and heterogeneous GNNs had the highest required training times (up to 13,039.06 s) compared to all other considered algorithms.

Afterward, we compared the robustness of algorithms against 10 and 100 noisy features (Fig 2, S3 Table). The ensemble-based machine learning algorithms (Random Forest, RUSBoost, XGBoost) and Logistic Regression required more training time (2.74 s to 809.45 s) but had nearly the same performance (only up to 0.0184 worse AUROC) compared to the original datasets. However, the Decision Tree (up to 0.1487 worse AUROC), the neural network (up to 0.1154 worse AUROC) and GNNs (up to 0.0804 worse AUROC) lost performance which indicates a high sensitivity against noise.

### 3.2 Partial dependence of graph and machine learning algorithms

After evaluating the performance, we evaluated partial dependence of each feature under the assumption of independent features. We plotted the average prediction (i.e., ratio of diseased sepsis-cases) against different feature values (lowest

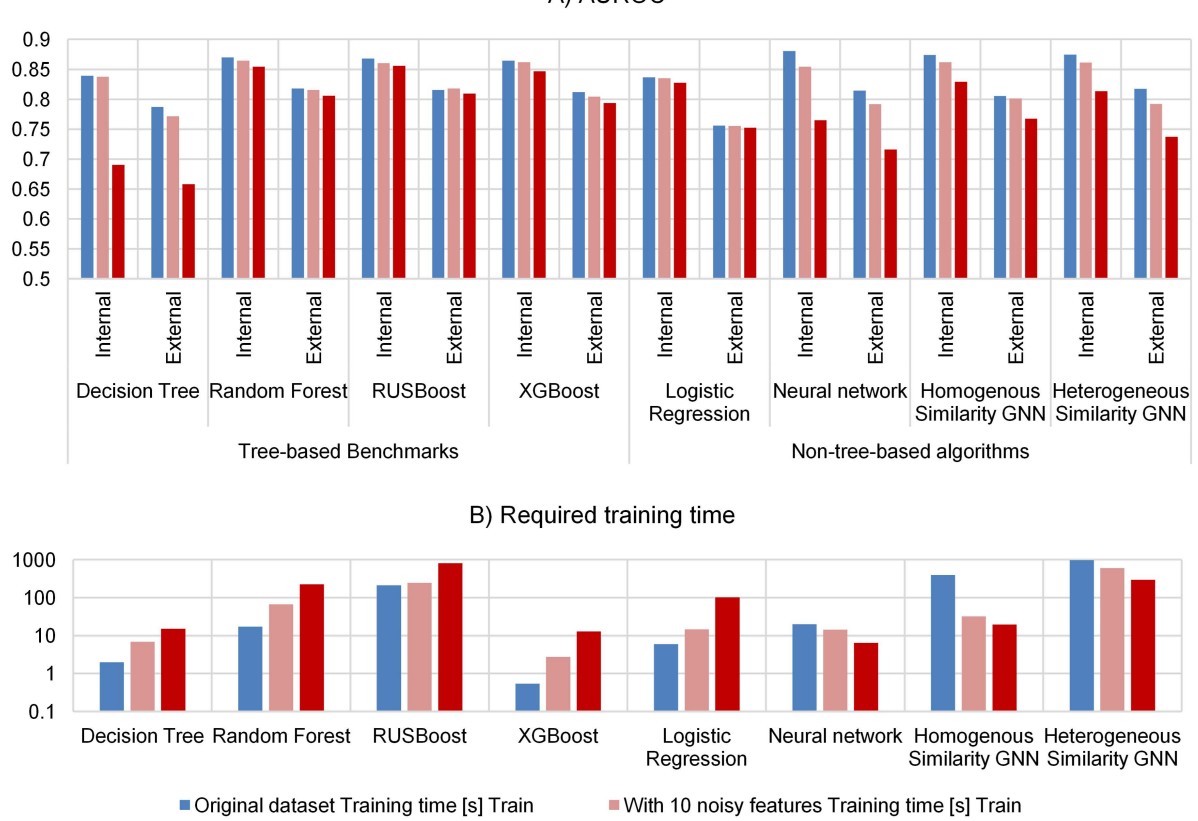

**Fig 2. Evaluating the robustness of GNNs compared to benchmarks (A) and their required training time (B).** We evaluated the classification performance (A) of GNNs (Graph-SAGE) on homogeneous and heterogeneous similarity graphs and the performance of other machine learning algorithms (neural network, Decision Tree, Logistic Regression, Random Forest, RUSBoost, XGBoost) by adding 10 or 100 noisy features to the complete blood count datasets (higher values represent better performance). Each model was trained on the noisy datasets and afterwards their classification performance was evaluated. Furthermore, we measured the required training time (B). We evaluated the performance on two datasets (internal and external dataset). Information about other evaluation metrics (F1-Score and MCC) are listed in S3 Table.

   

to highest value) (Fig 3A–F) for the features age (Fig 3A), hemoglobin (Fig 3C), red blood cells (Fig 3E), white blood cells (Fig 3D), mean corpuscular volume (Fig 3F) and platelets (Fig 3G). If the resulting curve has a positive gradient, an increasing feature value (e.g., older people for the feature age) results in an increased probability of developing sepsis according to the model, and vice versa for a negative gradient. Overall, there was a positive gradient for the features age, white blood cells (Fig 3D), red blood cells (Fig 3E), and mean corpuscular volume (Fig 3F), showing increased sepsis risk with rising values of these parameters. Specifically, that means older people with higher white blood cell counts, red blood cell counts and increased corpuscular volume have higher sepsis probabilities according to the trained models. White blood cells had a minimum of around 4–8 Gpt/l (gigaparticles per liter) for most algorithms (Fig 3D) that indicates a physiological white blood cell count at this range. In contrast to the positive gradient, we observed a negative gradient for platelets for all algorithms (Fig 3G), indicating decreased sepsis risk for rising values according to the models. Specifically, this means that patients with lower platelets (thrombocytopenia) have a higher sepsis probability according to the trained models. Tree-based algorithms (Decision Tree, RUSBoost, Random Forest, XGBoost) do not depend (gradient near zero) on the features hemoglobin (Fig 3C) and red blood cells (Fig 3E) in contrast to non-tree-based algorithms (Logistic Regression, Neural Network, Homogeneous GNN, Heterogeneous GNN). The latter ones showed an increased sepsis probability for low hemoglobin levels (anemia) (Fig 3C). Finally, the feature "sex" is nearly irrelevant for all algorithms (i.e., low gradient), i.e., the sepsis probability does not significantly depend on sex (Fig 3B).

The partial dependence plots (Fig 3A–G) contain the ratios of diseased cases for different feature values. We estimated the importance of each feature by calculating the variance of all plotted ratios for each feature. This feature importance is normalized over the sum of all features' importance in the model to obtain values between zero and one which sum up to one for all features (Fig 3H). For nearly all algorithms (besides Logistic Regression, and homogeneous GNN) white blood cell count was the most important feature. Furthermore, tree-based algorithms mainly rely on white blood cell count for their prediction. In contrast, the feature "sex" has the lowest feature importance across all models.

### 3.3 Graph neural networks on patient-centric graphs

Since the similarity graphs do not consider multiple measurements of the same patient (Fig 4A-C), we created patient-centric graphs and applied graph attention networks on them (Table 2). The graph attention networks sample, weight, and aggregate information from other measurements of the same patient either from previous (directed patient-graphs), subsequent (reversed directed patient-graphs) or all measurements (undirected patient graphs). The reversed directed patient-graphs (Fig 4E, Table 2) with and without positional encodings achieved the highest performance on both datasets, with an AUROC of up to 0.9565. The undirected graph (Fig 4F, Table 2) also achieved a superior performance with an AUROC of up to 0.9094. The directed graph (Fig 4D, Table 2) only achieved an AUROC of up to 0.8902. However, the performance without any positional encodings (Table 2) was lower with an AUROC of, 0.9502, 0.8996 and 0.8734 on the internal dataset for the reversed directed graph, the undirected graph and the directed graph, respectively.

To evaluate the contribution of the attention mechanism, we evaluated the classification performance on two other GNNs, GraphSAGE and GCN (Table 3). GraphSAGE achieved a slightly higher classification performance on all graph structures but is still comparable to the results of the GAT on both datasets. However, GCN performed slightly worse on the directed and undirected graph structure, but still achieved a similar performance on the reversed directed graph on both datasets.

Other deep learning techniques (LSTM, Bi-LSTM, 1D-CNN and Transformer) highly differed in their classification performance with an AUROC between 0.8802 and 0.9535 on the internal dataset (Table 4). The performance of the LSTM (AUROC on the internal dataset: 0.8802) was comparable or slightly worse to the GNN on directed patient-centric graphs (AUROC up to 0.9001). However, the Bi-LSTM (AUROC on the internal dataset: 0.9535) achieved comparable results to the reversed patient-centric graphs (AUROC up to 0.9575). The 1D-CNN (AUROC on the internal dataset: 0.9337) and the Transformer (AUROC on the internal dataset: 0.9257) had a slightly lower AUROC.

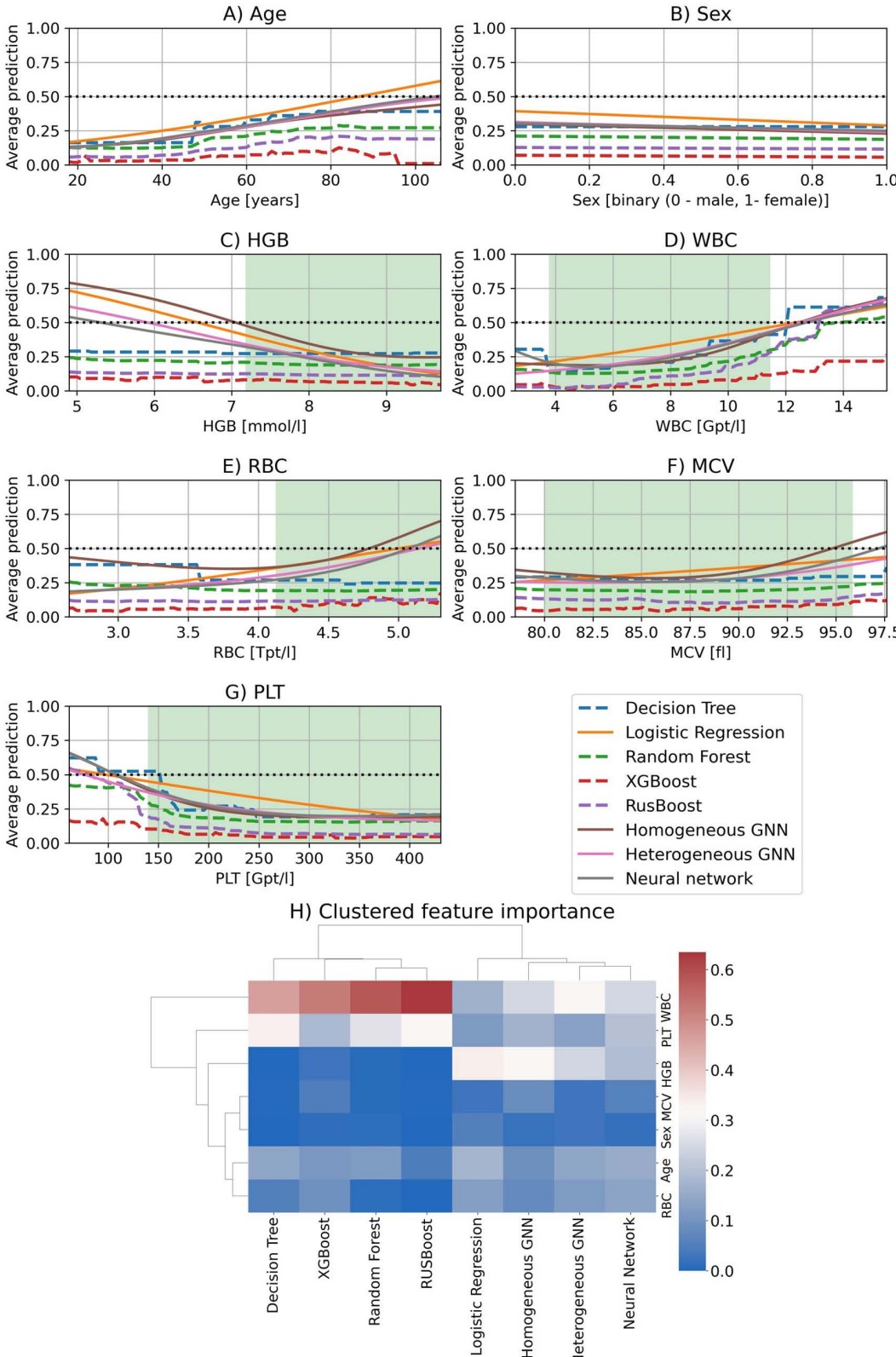

**Fig 3. Partial dependence plots (A – G) and the resulting clustered feature importance (H) for each feature and trained model.** We plotted the partial dependence plots for the features age in years (A), binary sex categorically encoded with one for women and zero for men (B), hemoglobin in

millimole per liter (C), white blood cells in gigaparticles per liter (D), red blood cells in teraparticles per liter (E), mean corpuscular volume in femtoliter (F), and platelets in gigaparticles per liter (G). For each feature, we plotted the average predictions (average ratio of sepsis classification) made by the trained models across different feature values (i.e., grid values). Tree-based algorithms (i.e., Decision Tree, Random Forest, XGBoost, and RUSBoost) are visualized as dashed lines and non-tree-based algorithms (i.e., Logistic Regression, the neural network, the homogeneous GNN, and heterogeneous GNN) as solid lines. If the curve in the feature variation graph has a positive gradient, an increasing feature value (e.g., older people for age) results in an increased probability of developing sepsis according to the model, and vice versa (negative gradient). Note, that at 0.5 (black dotted line) there is an equal number of sepsis and control cases according to the model. Therefore, we cannot make a statement around 0.5. Furthermore, we highlighted the reference range of specific blood parameters with a green rectangle [56,57] (see S4 Table). In H), we hierarchically clustered (Euclidean distance with average linking) the feature importance resulting from the normalized variance in the partial dependence plots for each trained model. Tree-based algorithms (i.e., Decision Tree, Random Forest, XGBoost, and RUSBoost) are grouped together indicating similar underlying mechanisms for the classification. However, their mechanisms differ from the non-tree-based algorithms (i.e., Logistic Regression, the neural network, the homogeneous GNN, and heterogeneous GNN) which are also grouped together.

## 4. Discussion and future directions

In this study, we evaluated the performance of GNNs on medical data using the use case sepsis prediction from blood count data. When GNNs are applied on similarity graphs (Table 1), they achieve similar performance as ensemble-based machine learning algorithms (XGBoost, RUSBoost, Random Forest) and neural networks. GNNs are based on the message-passing framework, i.e., information from connected nodes is iteratively transformed and aggregated to update node embeddings. There are only differences in the transformation and aggregation process [14]. GraphSAGE is only based on a linear transformation, followed by a mean aggregation of neighboring nodes [7]. GAT [8] and GATv2 [25] are also based on a linear transformation but integrate an attention mechanism to aggregate the information from neighboring nodes by a weighted (based on the neighbor's importance) mean. However, GATv2 has a more expressive attention mechanism by applying the attention score after the non-linearity (leaky rectified linear unit activation) instead of before [25]. Graph Isomorphism Network aim to increase the expressive power of GNNs by first summing information from neighboring nodes and then passing this aggregated information to a multi-layer perceptron to update the node embeddings [10]. Heterogeneous Graph Attention Network is using a node-level attention mechanism specific for each edge type and then aggregating information from different edge types by a semantic-based attention [11]. Heterogeneous Graph Transformer performs an attention-mechanisms based on query, key, and value matrices followed by linear transformation and a Gaussian Error Linear Unit activation function and another linear transformation [12]. However, all GNNs aggregate information from connected nodes to improve the performance while sharing the weights across nodes in the same layer [14]. However, when they can only aggregate information from similar complete blood count data (measurement-centric graphs), there information gain is too small resulting in a similar performance compared to the neural network and ensemble-based machine learning algorithms. Furthermore, the performance of some GNNs can drop below (e.g., Heterogeneous Graph Attention Network and Heterogeneous Graph Transformer) due to the high number of introduced parameters leading to slight overfitting [11,12]. Nevertheless, GNNs on similarity graphs, the neural network and ensemble-based algorithms could outperform shallow algorithms (Decision Tree and Logistic Regression) due to more expressive representations of the underlying information (Table 1). However, this performance increase is associated with an increased computational complexity which requires more training time compared to shallow algorithms. The increased computational complexity can be compensated when exploiting modern hardware (e.g., usage of multiple threads or GPUs). Thereby, the training of XGBoost on the GPU (NVIDIA A6000) requires even less time than the training of shallow algorithms. GNNs were also trained on a GPU but still required much more training time due to the high computational complexity of the underlying sampling, transformation, and aggregation steps (see Introduction). In general, the computational complexity of a single convolution layer of a GNN is $O(VFF' + EF')$, where V represents the number of nodes in the graph, E the number of edges in the graph, F the number of input features and F' the hidden dimension or output features for the convolution [8,43]. However, the computational complexity differs across different

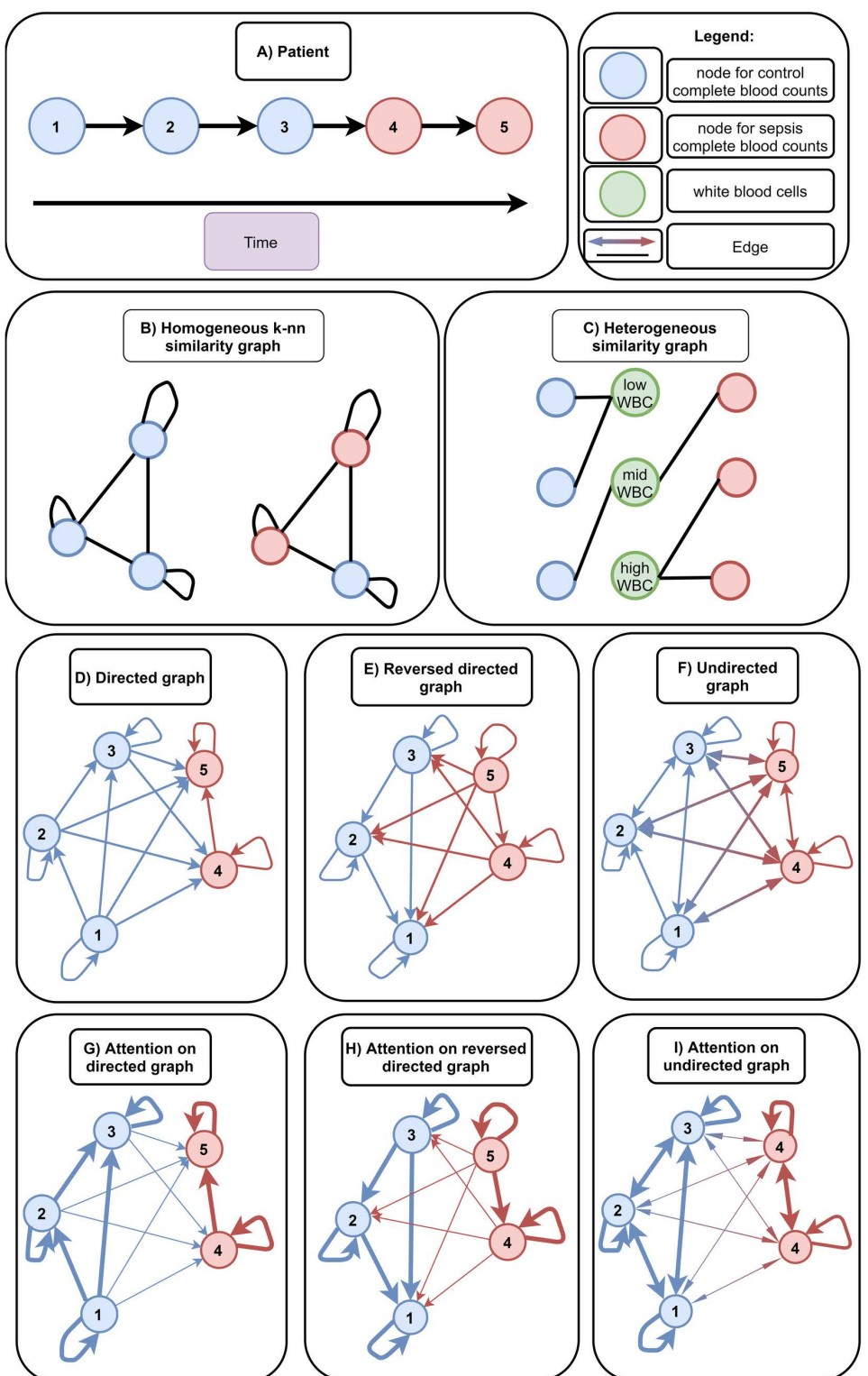

**Fig 4. Complete blood counts of a single patient** (A)**, design of used similarity graphs** (B, C)**, patient-centric graphs** (D-F) **and visualization of attention-weights/influence in patient-centric graphs after training** (G-I)**.** Each patient can have multiple blood count measurements ordered sequentially from 1 to 5 (A). Blood count measurement nodes labeled as "control" are highlighted in blue and measurements labeled as "sepsis" are

highlighted in red. First, we constructed two similarity graphs, a homogeneous k-nn graph (B) and a heterogeneous graph (C). Note, that the similarity graphs can comprise measurements from different patients depending on their feature values. In the homogeneous similarity graph (B) the black edges (represented as lines) are constructed based on the k-nearest neighbors of each blood count measurement node. The k-nearest neighbors are constructed using the Euclidean distance of standard normalized sex, age, hemoglobin, red blood cells, white blood cells, mean corpuscular volume, and platelets. The heterogeneous similarity graph (C) contains five additional node types, one node type for each blood parameter (i.e., hemoglobin, red blood cells, white blood cells, mean corpuscular volume, and platelets). Note, that we have only visualized one additional node type for simplicity. Each blood parameter node type has m nodes, where m denotes the number of percentiles we want to consider (in this example m = 3). Furthermore, each blood parameter node contains three features (i.e., minimum and maximum absolute value of the respective percentile interval and the upper percentile value) (not visualized). In this example, we divided the blood parameter white blood cells into three non-overlapping percentiles (i.e., the lowest 33.33%, 33.33% − 66.67%, and finally 66.67% −100% of the lowest values). Then, each blood count node is connected to the respective interval for each blood parameter. Thereby, blood count nodes are indirectly connected based on similarity. Since the similarity graphs do not consider multiple measurements of the same patient, we created patient-centric graphs (D-F). Based on the measurements of a single patient, we can construct a directed graph (D), reversed directed graph (E), and undirected graph (F). In the directed graph (D) measurements are connected with all previous measurements. The reversed directed graph (E) connects all following measurements to the present measurement. Finally, the undirected graph connects all measurements with each other independent from their order. Edges are colored based on the label of the source node. Afterwards, we trained graph attention networks (GAT) on these graphs either with or without positional encodings (to represent their order in the sequence). GAT employs an attention mechanism which finally leads to a weighted sum (attention) of a neighbors' measurements. We have schematically visualized the attention weights (i.e., their weight/influence on the target node) by the edge thickness for each graph (G-I). After training, nodes targeting nodes with the same label (e.g., control node to control node) have relatively high influences (thick edge), while nodes targeting nodes with a different label (e.g., sepsis node to control node) have relatively small influences (thin edges).

**Table 2. Comparing AUROC, exploited biases, use cases and issues of Graph Attention Networks on similarity and patient-centric graphs (directed, reversed directed and undirected) for the classification of sepsis on complete blood count data (higher values represent better performance). We evaluated the classification performance on two datasets (internal and external dataset). Bold values represent the best values in each column.**

| Graph | | AUROC | | Structure bias | Feature bias | Use case | Issue |
|---|---|---|---|---|---|---|---|
| | | Internal | External | | | | |
| Measurement-centric graphs | Homogeneous similarity graph (Fig 4B) | 0.8741 | 0.8052 | − | − | Time-specific diagnostic (e.g., to predict sepsis) | Does the patient have sepsis according to the available information? |
| | Heterogeneous similarity graph (Fig 4C) | 0.8747 | 0.8176 | − | − | | |
| Patient-centric graphs | Directed graph (Fig 4D) | 0.8734 | 0.8114 | − | − | | |
| | Directed graph with positional encodings (Fig 4D) | 0.8902 | 0.8203 | − | + | | |
| | Undirected graph (Fig 4F) | 0.8996 | 0.8628 | − | − | Retrospective diagnostics (e.g., for evaluating treatment strategies or disease causes) | When did the patient diseased or recovered? |
| | Undirected graph with positional encodings (Fig 4F) | 0.9094 | 0.8652 | − | + | | |
| | Reversed directed graph (Fig 4E) | 0.9502 | 0.9481 | + | − | | |
| | Reversed directed graph with positional encodings (Fig 4E) | **0.9565** | **0.9498** | + | + | | |

architectures, depending on factors like the number of attention heads (e.g., in GAT and GATv2), and used multi-layer perceptrons (e.g., in GIN).

In addition to computational time, tree-based ensemble algorithms (XGBoost, RUSBoost, Random Forest) are more robust against noise compared to the Decision Tree, GNNs and the neural network (Fig 2). The increased robustness might be the result of the aggregation of multiple tree-based algorithms (ensembles). It is noteworthy, that the neural

**Table 3. Comparing AUROC across Graph Neural Networks patient-centric graphs (directed, reversed directed and undirected) for the classification of sepsis on complete blood count data (higher values represent better performance). We evaluated the classification performance on two datasets (internal and external dataset). Bold values represent the best values in each column.**

| Graph | | GAT | | GraphSAGE | | GCN | |
|---|---|---|---|---|---|---|---|
| | | Internal | External | Internal | External | Internal | External |
| **Patient-centric graphs** | Directed graph (Fig 4D) | 0.8734 | 0.8114 | 0.8890 | 0.8323 | 0.8582 | 0.7871 |
| | Directed graph with positional encodings (Fig 4D) | 0.8902 | 0.8203 | 0.9001 | 0.8338 | 0.8613 | 0.7900 |
| | Undirected graph (Fig 4F) | 0.8996 | 0.8628 | 0.9174 | 0.8831 | 0.8742 | 0.8083 |
| | Undirected graph with positional encodings (Fig 4F) | 0.9094 | 0.8652 | 0.9336 | 0.9124 | 0.8955 | 0.8449 |
| | Reversed directed graph (Fig 4E) | 0.9502 | 0.9481 | 0.9542 | 0.9499 | **0.9575** | **0.9520** |
| | Reversed directed graph with positional encodings (Fig 4E) | **0.9565** | **0.9498** | **0.9545** | **0.9514** | 0.9517 | 0.9502 |

**Table 4. Comparing AUROC across several deep learning techniques for the classification of sepsis on complete blood count data (higher values represent better performance). We evaluated the classification performance on two datasets (internal and external dataset). Bold values represent the best values in each column.**

| Graph | Patient-centric graphs | | Use case |
|---|---|---|---|
| | Internal | External | |
| **LSTM** | 0.8802 | 0.8206 | Time-specific diagnostic (e.g., to predict sepsis) |
| **Bi-LSTM** | **0.9535** | **0.9557** | Retrospective diagnostics (e.g., for evaluating treatment strategies or disease causes) |
| **1D-CNN** | 0.9337 | 0.9408 | |
| **Transformer** | 0.9257 | 0.9208 | |

network and GNN required less training time with more noisy features which is due to the faster convergence to new (but worse) local minima while training.

Afterward, we evaluated the partial dependence and importance of different features for the final classification of each model (Fig 3A-G). Tree-based algorithms (Decision Tree, Random Forest, RUSBoost, XGBoost) showed similar partial dependence plots for classification which results from a similar prediction mechanism (usage of a single or multiple Decision Tree(s)). However, these mechanisms differ from non-tree-based algorithms (Logistic Regression, neural networks, GNNs) since they are based on linear transformation with (neural network and GNNs) or without (Logistic Regression) some kind of non-linearity (e.g., sigmoid, or rectified linear unit). Additionally, tree-based algorithms create harder decision boundaries than non-tree-based algorithms. Our approach for increasing the interpretability of machine learning models assumes that all features are independent from each other. However, in reality features are dependent on each other (e.g., red blood cells and hemoglobin). This simplification might skew the synthetic feature inputs for specific combinations. Future approaches could integrate existent feature dependencies to prevent distortions in the synthetic dataset.

Finally, we tested the performance of Graph Attention Networks on patient-centric graphs (i.e., graphs which integrate measurements of the same patient) (Table 2). The exploitation of time series information through the patient-centric graphs improved the classification performance of all previous models and achieved an AUROC of up to 0.9565 on Graph Attention Networks. The reason for this improvement is that the GNN on a patient-centric graph inherently reduces patient-specific fluctuations in the dataset. However, the performance improvement is also associated with the exploitation of a real-world bias in the underlying dataset. About 2/3 of the sepsis cases are not part of a sequence of examinations (i.e., they represent only a single measurement for a patient). However, the other 1/3 of the sepsis cases are part of examination sequences and sepsis is diagnosed in the last positions in most cases (92.14%). This highlights the benefit of regular monitoring of patient data as baseline information for machine learning algorithms.

The fact that most sepsis cases occur only at the last positions is exploited by biased features attributes (feature-induced bias) and/or a biased underlying graph structure (structure-induced bias). When incorporating positional encodings, we represent later positions (i.e., measurements) with higher feature attributes and earlier ones with lower feature attributes (feature-induced bias). With a specific graph structure (reversed directed, Fig 4E), the underrepresented sepsis cases do not integrate feature information from control information (structure-induced bias). However, the control cases can share information between each other which reduces potential fluctuations. Although the control cases can also integrate information from sepsis cases, the attention mechanism reduces their influence (Fig 4H, S5 Table). In the directed and undirected graph, control cases still share information with each other to reduce potential fluctuations. However, sepsis cases also integrate information from control cases which reduces the differences between the two groups. The integration of information from control cases is partially compensated by the attention mechanism which lowers the influence of control cases to sepsis cases (Fig 4G and I, S5 Table) but cannot be fully compensated due to the high number of control cases in contrast to sepsis cases. Thereby, the reversed directed patient-graphs achieve a much higher classification performance (AUROC of up to 0.9565) compared to the directed and (AUROC of up to 0.9094) and undirected graphs (AUROC of up to 0.8902).

Furthermore, we evaluated the performance on other GNNs (GraphSAGE [7] and GCN [9]) to evaluate whether the improvement was the consequence of the underlying attention mechanism (Table 3). The similar classification performance of GraphSAGE indicates that the attention mechanism is not required to achieve the improvement. A slightly higher AUROC could be the result of a less complex GNN structure (only a linear transformation of aggregated nodes without attention mechanism). Furthermore, GraphSAGE was developed to increase the inductive capabilities of GNNs [7]. GCN achieved a slightly lower AUROC on the directed and undirected graph structures which might be the result of the symmetrical normalization and the linear transformation of data before the aggregation [9]. An aggregation before linear transformation (e.g., in GraphSAGE) might smooth the node features facilitating the classification with the following linear transformation.

Compared to other deep learning techniques, GNNs achieved similar or even higher classification performance (AUROC) (Table 4). The LSTM [40] achieved similar performance to GNNs on directed patient-centric graphs. Similar to the GNNs on directed patient-centric graphs, the LSTM can only process information from past observations (i.e., complete blood count data) limiting the information for complete blood count at the beginning of time series. However, the Bi-LSTM [41], 1D-CNN [44], and Transformer [13] can aggregate information independently of their order (i.e., future and past observations) increasing the available information for all complete blood count samples. Thereby, the AUROC increased up to 0.9535 on the internal dataset and achieved similar results to the best performing GNN (AUROC up to 0.9575). The 1D-CNN and Transformer might performed slightly worse because they incorporate information from both directions into one hidden representation, while the Bi-LSTM creates two separate hidden representations (one for the forward direction and one for the backward direction) which are combined at the end. Furthermore, it is noteworthy that GNNs can handle time series of different lengths natively by the defined edge index while the other deep learning architectures required an additional padding and masking step.

We can use undirected and reversed directed patient-graphs for retrospective analysis (e.g., when a patient is diseased or recovered). This application might help to evaluate the success of a treatment (e.g., with specific antibiotics) or to evaluate potential causes of a disease (e.g., infection after a specific event). However, we cannot use the undirected and reversed directed patient-graphs when we want to diagnose sepsis at the current time point since they are incorporating information from subsequent measurements (i.e., information not available at the current time point). The same holds for the Bi-LSTM, 1D-CNN and Transformer. Therefore, we can only use the directed patient-centric graphs with and without positional encodings or the LSTM which achieved a lower classification performance compared to GAT on the undirected and reverse directed graphs. However, the performance of the directed patient-centric graph with positional encodings (AUROC of up to 0.8902) is still better than the standard ML algorithms (AUROC of up to 0.8806) which did not use time-series information.

To sum up, we compared the classification performance of different graph learning and other machine learning algorithms on sepsis blood count data and revealed different classification mechanisms in the trained models. Furthermore, we evaluated the performance of Graph Attention Networks on several patient-centric graphs and reached an outstanding AUROC of up to 0.9565 for retrospective use cases.

We would suggest the following directions for future research:

I. Integrating additional features;

II. Integrating more samples;

III. Diagnoses of further diseases.

The integration of more features (I.) could include information of other laboratory measurements (e.g., specific biomarkers), vital signs of patients (e.g., body temperature and pulse rate), predisposing factors (e.g., genetic polymorphism [45] or chronic medical conditions like diabetes [46]), and previously administered drugs. These features might help to provide a more holistic view of a patient's health status. Furthermore, sparse information like the existence of predisposing factors or previously administered drugs could be represented as a graph structure. However, data with more features must be collected for all patients, which could increase measurement times and costs. Furthermore, specific information like administered drugs could contain information clinicians might have only in retrospect. The integration of more samples (II.) in the dataset is time-consuming but could reduce the impact of outliers in the dataset. One promising direction might be the integration of samples from electronic health records like MIMIC-IV [47], Amsterdam University Medical Center Database [48], high time resolution ICU dataset (HiRID) [49] and eICU Collaborative Research Database [50]. Additionally, complete count data could enable diagnosing further diseases (III.) like thrombosis [51] or leukemia [52]. For the classification of further diseases, labels for the respective diseases are required. However, this labeling process might be time-consuming and requires domain experts like clinicians.

## Supporting information

**S1 Note. Potential relevance of complete blood count data to sepsis**.
(DOCX)

**S2 Fig. Distribution of each continuous feature in the train, internal and external test datasets as violin plots.**
(DOCX)

**S3 Table. Evaluating the robustness of GNNs compared to benchmarks by adding 10 or 100 noisy features to the complete blood count datasets for sepsis classification (higher values represent better performance) and the required training time.**
(DOCX)

**S4 Table. Reference values for blood parameters in complete blood count analysis.**
(DOCX)

**S5 Table. Attention weights of the last layer of the trained graph attention networks on each graph (directed graph, reverse directed graph and undirected graph).**
(DOCX)

## Author contributions

**Conceptualization:** Daniel Walke, Daniel Steinbach, Sebastian Gibb, Thorsten Kaiser, Paul C. Ahrens, Gunter Saake, David Broneske, Robert Heyer.

**Data curation:** Daniel Steinbach, Sebastian Gibb, Thorsten Kaiser, Paul C. Ahrens.

**Formal analysis:** Daniel Walke.

**Funding acquisition:** Gunter Saake, David Broneske, Robert Heyer.

**Investigation:** Daniel Walke.

**Methodology:** Daniel Walke.

**Project administration:** David Broneske, Robert Heyer.

**Software:** Daniel Walke.

**Supervision:** Daniel Steinbach, Sebastian Gibb, Thorsten Kaiser, Gunter Saake, David Broneske, Robert Heyer.

**Visualization:** Daniel Walke.

**Writing – original draft:** Daniel Walke.

**Writing – review & editing:** Daniel Walke, Daniel Steinbach, Sebastian Gibb, Thorsten Kaiser, Paul C. Ahrens, Gunter Saake, David Broneske, Robert Heyer.

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
