## [Decision Letter · Decision Letter 0]

Dear Dr. Walke,

Thank you for submitting your manuscript to PLOS ONE. After careful consideration, we feel that it has merit but does not fully meet PLOS ONE’s publication criteria as it currently stands. Therefore, we invite you to submit a revised version of the manuscript that addresses the points raised during the review process.

We look forward to receiving your revised manuscript.

Kind regards,

Hanna Landenmark

Staff Editor

PLOS ONE

“We thank all co-authors for contributing to the manuscript. Furthermore, we thank the German Research Foundation (DFG) for funding this project [grant numbers HE 8077/2-1, SA 465/53-1].”

“Grant-Numbers: HE 8077/2-1, SA 465/53-1

Funder: German Research Foundation/ Deutsche Forschungsgemeinschaft (DFG) (https://www.dfg.de/de)

Funded Authors: D. W.

Reviewers' comments:

Reviewer's Responses to Questions

**Comments to the Author**

1. Is the manuscript technically sound, and do the data support the conclusions?

Reviewer #1: No

Reviewer #2: Partly

2. Has the statistical analysis been performed appropriately and rigorously?

Reviewer #1: Yes

Reviewer #2: No

3. Have the authors made all data underlying the findings in their manuscript fully available?

Reviewer #1: Yes

Reviewer #2: Yes

4. Is the manuscript presented in an intelligible fashion and written in standard English?

Reviewer #1: Yes

Reviewer #2: Yes

Reviewer #1: The authors emphasize the important of making connections between samples.

However, the experimental results do not show that it is more effective to analyze samples from the perspective of graphs.

There are some concerns.

When analyzing similar-graphs, GNNs are comparable in performance to other algorithms, but less robust and longer training time;

When analyzing patient-centric graphs, it seems more like the attention mechanism works, not graph struture. I do not think other GNNs (i.e. GCN) can achieve the same high performance. More experiments may be needed here.

In addition, I wonder how the attributes of various types of nodes in heterogeneous similarity graphs are defined, since the attributes in homogeneous similarity graphs are used as nodes.

Despite these concerns, I think it makes sense to use GNNs to analyze medical data. The authors may need to further explore the way they consturct the graphs, any why GNNs are more efficient.

Reviewer #2: This manuscript titled “Edges are all you need: Potential of Medical Time Series 2 Analysis with Graph Neural Networks” introduced an approach for incorporating time-series clinical diagnosis data efficiently, and showed that graph neural networks (GNNs) provide a better alternative than traditional machine learning (ML) algorithms. While I think this work is very well-written and well explained, the novelty and impact of this work falls short for acceptance in PLOS One. Therefore, I reject this manuscript. I addressed the concerns below:

1. Novelty issue: The main objective of this paper and the graph neural network based approach is not new for clinical data. Several previous works have already applied GNN on clinical data. Some of the previous works are listed below [1 - 4]. Moreover, the feature importance calculation part is also not the invention of the authors, as their mentioned process falls under a special form of ablation study highly done in GNN papers. Furthermore, the GNNs they used to evaluate performance are also not developed by the authors. Under these circumstances, I believe this work has not been able to meet the novelty criteria for PLOS One.

[1] Wang, Yanan, Yu Guang Wang, Changyuan Hu, Ming Li, Yanan Fan, Nina Otter, Ikuan Sam et al. "Cell graph neural networks enable the precise prediction of patient survival in gastric cancer." NPJ precision oncology 6, no. 1 (2022): 45.

[2] Sun, Zhenchao, Hongzhi Yin, Hongxu Chen, Tong Chen, Lizhen Cui, and Fan Yang. "Disease prediction via graph neural networks." IEEE Journal of Biomedical and Health Informatics 25, no. 3 (2020): 818-826.

[3] Gao, Jianliang, Tengfei Lyu, Fan Xiong, Jianxin Wang, Weimao Ke, and Zhao Li. "Predicting the survival of cancer patients with multimodal graph neural network." IEEE/ACM Transactions on Computational Biology and Bioinformatics 19, no. 2 (2021): 699-709.

[4] Li, Yang, Buyue Qian, Xianli Zhang, and Hui Liu. "Graph neural network-based diagnosis prediction." Big data 8, no. 5 (2020): 379-390.

2. Work amount issue: The authors only evaluated performances for one dataset which is quite inadequate for evaluating performances across different types of data. Without showing that their approach generalizes for multiple types of data, this work cannot be accepted.

3. Results issue: According to the authors comments, if incorporating time series data improves performance for these types of data, than better alternative are recurrent neural networks (RNNs) and Transformers. But no comparison was shown of the GNNs with these types of models. GNNs are more suited for data that has natural graph-like structures (e.g., crystals, proteins, molecules, RNA, etc.). So, without comparing it with at least an LSTM model (RNN) [1], I cannot understand the impact of this work.

[1] Hochreiter, Sepp, and Jürgen Schmidhuber. "Long short-term memory." Neural computation 9, no. 8 (1997): 1735-1780.

4. “However, this information is mostly neglected by state-of-the-art machine learning algorithms" - you need to cite some works.

5. “Furthermore, GNNs and other complex machine learning algorithms (e.g., XGBoost) are often treated as black-boxes limiting their interpretability and transparency which is essential for medical applications.” - this work also does not address this issue. The feature importance calculation does not address this issue as this refers to the interpretability of the neural network itself. For example, what sort of information the output of each layer (latent space) bears.

6. What is the validity of synthetic data generated? No explanation was provided.

7. “The reason for this similar performance is that the nodes of complete blood counts only sample information from similar node blood count measurements (measurement-centric graphs).” - not a strong reason, need to describe with respect to the GNN architecture.

8. No details on the GNN algorithms used. The readers need to know the scientific reasons why GNN is performing better than traditional ML models. The authors need to explain why particular GNN architecture performed better, and why particular GNN architecture performed worse. Because this work can be iteratively improved, but if they don’t delve into the GNN architecture, this becomes very hard to improve logically.

**Do you want your identity to be public for this peer review?** For information about this choice, including consent withdrawal, please see our Privacy Policy

Reviewer #1: No

Reviewer #2: No

---

## [Author Response · Author response to Decision Letter 1]

24 Jun 2024

Reviewer 1

Thank you for your feedback. We hope that the extended results improved our manuscript.

The authors emphasize the important of making connections between samples.

However, the experimental results do not show that it is more effective to analyze samples from the perspective of graphs. There are some concerns.

#R1_1: When analyzing similar-graphs, GNNs are comparable in performance to other algorithms, but less robust and longer training time;

#R1_1 Answer: Yes, that is why we would like to emphasize not using GNNs for similarity-based graph structures. However, we can achieve improvements with GNNs on patient-centric graphs (see #R1_2).

#R1_2: When analyzing patient-centric graphs, it seems more like the attention mechanism works, not graph struture. I do not think other GNNs (i.e. GCN) can achieve the same high performance. More experiments may be needed here.

#R1_2 Answer: We performed further experiments of GNNs (GraphSAGE and GCN) on patient-centric graphs. The performance on GraphSAGE was comparable to the performance of GAT indicating that the attention mechanism is not necessary for the observed improvements. GCN has a bit lower performance on directed and undirected graph structures which might be the consequence of the symmetric normalization and the linear transformation before the message-propagation (e.g., GraphSAGE first propagate and then transforms the data). However, on the reversed-directed graphs the performance was similar to the GAT (see #R1_2).

#R1_3: In addition, I wonder how the attributes of various types of nodes in heterogeneous similarity graphs are defined, since the attributes in homogeneous similarity graphs are used as nodes.

#R1_3 Answer: We tried to clarify the explanation more (see #R1_3).

#R1_4: Despite these concerns, I think it makes sense to use GNNs to analyze medical data. The authors may need to further explore the way they consturct the graphs, any why GNNs are more efficient.

#R1_4 Answer: Our main objective was to improve clinical decision support systems with clinical routine data like complete blood count data with algorithms best suited for the underlying data. We highlighted this focus now better in our title. In this process, we figured out that incorporating the information from time series information (patient-centric graphs) was much more promising than considering the complete blood count measurements independent of the patient (e.g., similarity graphs).

Reviewer 2

Thank you for your comprehensive feedback on our work. We hope that the updated changes and extended results improved the manuscript.

This manuscript titled “Edges are all you need: Potential of Medical Time Series 2 Analysis with Graph Neural Networks” introduced an approach for incorporating time-series clinical diagnosis data efficiently, and showed that graph neural networks (GNNs) provide a better alternative than traditional machine learning (ML) algorithms. While I think this work is very well-written and well explained, the novelty and impact of this work falls short for acceptance in PLOS One. Therefore, I reject this manuscript. I addressed the concerns below:

#R2_1: 1. Novelty issue: The main objective of this paper and the graph neural network based approach is not new for clinical data. Several previous works have already applied GNN on clinical data. Some of the previous works are listed below [1 - 4]. Moreover, the feature importance calculation part is also not the invention of the authors, as their mentioned process falls under a special form of ablation study highly done in GNN papers. Furthermore, the GNNs they used to evaluate performance are also not developed by the authors. Under these circumstances, I believe this work has not been able to meet the novelty criteria for PLOS One.

[1] Wang, Yanan, Yu Guang Wang, Changyuan Hu, Ming Li, Yanan Fan, Nina Otter, Ikuan Sam et al. "Cell graph neural networks enable the precise prediction of patient survival in gastric cancer." NPJ precision oncology 6, no. 1 (2022): 45.

[2] Sun, Zhenchao, Hongzhi Yin, Hongxu Chen, Tong Chen, Lizhen Cui, and Fan Yang. "Disease prediction via graph neural networks." IEEE Journal of Biomedical and Health Informatics 25, no. 3 (2020): 818-826.

[3] Gao, Jianliang, Tengfei Lyu, Fan Xiong, Jianxin Wang, Weimao Ke, and Zhao Li. "Predicting the survival of cancer patients with multimodal graph neural network." IEEE/ACM Transactions on Computational Biology and Bioinformatics 19, no. 2 (2021): 699-709.

[4] Li, Yang, Buyue Qian, Xianli Zhang, and Hui Liu. "Graph neural network-based diagnosis prediction." Big data 8, no. 5 (2020): 379-390.

#R2_1 Answer: We included the references in our introduction section and have rewritten the sections and clarified our main objectives (see #R2_1). Our main objective is to improve clinical decision support systems with clinical routine data like complete blood count data with algorithms best suited for the underlying data (time series data of different length). Regardless of this objective, the novelty criteria does not apply to PLOS ONE as indicated on their homepage: „The world’s first multidisciplinary Open Access journal, PLOS ONE accepts scientifically rigorous research, regardless of novelty.“ (see https://everyone.plos.org/about-plos-one/)

#R2_2: 2. Work amount issue: The authors only evaluated performances for one dataset which is quite inadequate for evaluating performances across different types of data. Without showing that their approach generalizes for multiple types of data, this work cannot be accepted.

#R2_2 Answer: We never intended nor described to evaluate the performance across multiple data types. Our objective was to evaluate the classification performance on complete blood count data from clinical routine (see answer #R2_1). An external validation of our results was provided with an external validation dataset from a different tertiary care center.

#R2_3 3. Results issue: According to the authors comments, if incorporating time series data improves performance for these types of data, than better alternative are recurrent neural networks (RNNs) and Transformers. But no comparison was shown of the GNNs with these types of models. GNNs are more suited for data that has natural graph-like structures (e.g., crystals, proteins, molecules, RNA, etc.). So, without comparing it with at least an LSTM model (RNN) [1], I cannot understand the impact of this work.

[1] Hochreiter, Sepp, and Jürgen Schmidhuber. "Long short-term memory." Neural computation 9, no. 8 (1997): 1735-1780.

#R2_3 Answer: We introduced an explicit decoder-only transformer architecture, 1D-CNN, LSTM and Bi-LSTM for the sake of completeness (see #R2_3). In contrast to the GNN architecture, these architectures (at least the Transformer and 1D-CNN, and for efficient batching also LSTM architectures) required an additional padding and masking step which exacerbates their usage for time series of different lengths.

#R2_4: “However, this information is mostly neglected by state-of-the-art machine learning algorithms" - you need to cite some works.

#R2_4 Answer: We clarified the statement (see #R2_4).

#R2_5 5. “Furthermore, GNNs and other complex machine learning algorithms (e.g., XGBoost) are often treated as black-boxes limiting their interpretability and transparency which is essential for medical applications.” - this work also does not address this issue. The feature importance calculation does not address this issue as this refers to the interpretability of the neural network itself. For example, what sort of information the output of each layer (latent space) bears.

#R2_6 6. What is the validity of synthetic data generated? No explanation was provided.

#R2_5+6 Answer: While writing the manuscript, we unfortunately were not aware of the existence of partial dependence plots (https://hastie.su.domains/ElemStatLearn/) which were precisely our goal (i.e., investigating prediction changes based on different feature values). Therefore, we changed our methodology to the partial dependence plots and plotted the average prediction values across different grid values between the 5% and 95% percentile (see #R2_5+6). Thereby, we can directly see the influence of different values of individual features to the overall prediction of a specific complete blood count measurement.

#R2_7+8: 7. “The reason for this similar performance is that the nodes of complete blood counts only sample information from similar node blood count measurements (measurement-centric graphs).” - not a strong reason, need to describe with respect to the GNN architecture.

8. No details on the GNN algorithms used. The readers need to know the scientific reasons why GNN is performing better than traditional ML models. The authors need to explain why particular GNN architecture performed better, and why particular GNN architecture performed worse. Because this work can be iteratively improved, but if they don’t delve into the GNN architecture, this becomes very hard to improve logically.

#R2_7+8 Answer: We added further information as explanations including the individual architectures of the investigated GNNs (see #R2_7+8)

---

## [Decision Letter · Decision Letter 1]

Dear Dr. Walke,

Thank you for submitting your manuscript to PLOS ONE. After careful consideration, we feel that it has merit but does not fully meet PLOS ONE’s publication criteria as it currently stands. Therefore, we invite you to submit a revised version of the manuscript that addresses the points raised during the review process.

We look forward to receiving your revised manuscript.

Kind regards,

Giacomo Fiumara, PhD

Academic Editor

PLOS ONE

Additional Editor Comments:

The second round of reviews is now complete. My opinion is that the manuscript must undergo a major revision before considering for publication in PLOS ONE.

What emerges is that the manuscript lacks a unitary style. In addition, the abstract and the introduction should (at least) mention all the algorithms used in the experiments. In this respect, the abstract and the introduction fail in presenting the research.

Reviewers' comments:

Reviewer's Responses to Questions

**Comments to the Author**

Reviewer #3: (No Response)

Reviewer #4: All comments have been addressed

2. Is the manuscript technically sound, and do the data support the conclusions?

Reviewer #3: Yes

Reviewer #4: No

3. Has the statistical analysis been performed appropriately and rigorously?

Reviewer #3: Yes

Reviewer #4: Yes

4. Have the authors made all data underlying the findings in their manuscript fully available?

Reviewer #3: Yes

Reviewer #4: Yes

5. Is the manuscript presented in an intelligible fashion and written in standard English?

Reviewer #3: Yes

Reviewer #4: Yes

Reviewer #3: The manuscript presents an original study that applies Graph Neural Networks (GNNs) to predict sepsis using Complete Blood Count (CBC) data. This is an interesting and promising approach, but the paper has several areas that require improvement. While the second half of the paper addresses many initial issues, there are gaps in clarity, the experimental setup, and interpretability that need to be tackled before the manuscript can be considered for publication.

Originality

The application of GNNs to patient-centric graphs for sepsis detection is novel and contributes to ongoing research into machine learning in healthcare. However, the work overlaps with previous studies like Chen et al. (Eur Rev Med Pharmacol Sci 2021; 25 (14): 4693-4701, DOI: 10.26355/eurrev_202107_26380), which used GNNs for pediatric sepsis. A more explicit comparison with this and similar works, along with a justification of the broader patient population, would strengthen the claim of originality.

Clinical Relevance

The selection of CBC parameters is reasonable, but the relevance of each feature to sepsis prediction should be better explained. For readers unfamiliar with clinical data, it would be helpful to briefly introduce why these specific markers are key indicators of sepsis and to support this with appropriate references.

Ethical Considerations

The manuscript states that ethical approval was not applicable due to the anonymization of the data. However, further elaboration is necessary regarding how the privacy of patient data was ensured. Data privacy is particularly sensitive in healthcare applications, and a brief explanation of how the dataset was anonymized would help meet ethical transparency standards.

Evaluation of Paper Sections

• Abstract

The abstract is concise but could be clearer. For instance, it mentions AUROC as the primary evaluation metric but doesn’t justify why this metric was chosen over others like the F1 score, which is more commonly use. A brief justification for focusing on AUROC would be helpful. Additionally, more details about the graph structure (why it was used over simpler models) and how GNNs handle time-series data would improve the clarity.

• Introduction

The introduction is somewhat lacking in depth, particularly concerning the clinical background of sepsis and the rationale for using GNNs. A more detailed explanation of why GNNs are suited for sepsis prediction, compared to simpler methods, would benefit the reader. The discussion of DeepWalk and Node2Vec compares them to GNNs, implying they are machine learning algorithms. These methods are more accurately described as graph embedding techniques used for feature representation, not prediction. This section would benefit from clarification and more technical precision. The introduction should also be expanded to give a clearer description of how the graph structure was defined. The definition of nodes and edges is somewhat imprecise and could confuse readers unfamiliar with graph theory terminology. For example, terms like "vertices" and "links" are interchangeable with "nodes" and "edges," but their use should be consistent.

• Methods

The methods section provides a detailed explanation of the experimental setup but lacks justification for some key decisions. For instance, why were tree-based algorithms chosen as baselines? Further, the reasoning behind certain hyperparameter choices, such as the number of epochs and learning rate, is unclear. More justification for selecting GraphSAGE as the representative GNN is necessary: were other GNN architectures considered, and if so, why was GraphSAGE chosen as the final model?

• Results

The results section is one of the paper’s strengths, providing a comprehensive evaluation of the models and a comparison between different machine learning approaches. However, the choice of AUROC as the primary metric remains problematic. The authors do mention F1 later in the paper but should provide more justification for emphasizing AUROC initially. The partial dependence plots in the results are well-executed, but the explanation of how the direction of individual features (e.g., an increase in white blood cell count) affects predictions could be clearer. While the paper mentions this limitation, providing more interpretability around feature impact is crucial, particularly in clinical settings where decisions depend on understanding how individual factors contribute to risk.

• Discussion

The discussion section does a good job of addressing the limitations of the study, including bias in the dataset and computational challenges. However, the paper would benefit from a more detailed analysis of the computational complexity of GNNs, which are known to be resource-intensive. Were any optimizations considered, such as using more efficient GNN architectures or limiting the number of layers to reduce computational overhead?

• Data and Code Availability

The transparency in making the code and data publicly available on GitHub and Zenodo is commendable and aligns with best practices in reproducibility. This adherence to open science is a strength of the paper, ensuring that the study can be replicated and verified by other researchers.

• Appendix

The appendix provides valuable supplementary materials, including tables, plots, and diagrams that enhance the main text. These materials clarify several points that are less detailed in the main sections.

Recommendation

The study presents a novel application of GNNs to sepsis prediction and shows strong experimental work. However, the manuscript requires revisions to improve its clarity and justification of choices in both methodology and evaluation metrics. I recommend a minor revision to address these issues before it can be considered for publication in PLOS ONE.

Reviewer #4: The article lacks coherence between the introduction and the conclusion and does not delve into the GNN algorithms used. It also fails to provide an explanation of why some GNN algorithms perform better than others, nor does it discuss the implemented architecture. This significantly reduces its scientific value. The absence of such crucial details makes it difficult for the reader to fully understand the work. Due to these reasons, as well as those outlined in the major comments (see attached file), I believe the article is not suitable for publication in the journal.

**Do you want your identity to be public for this peer review?** For information about this choice, including consent withdrawal, please see our Privacy Policy

Reviewer #3: No

Reviewer #4: No

---

## [Author Response · Author response to Decision Letter 2]

27 Nov 2024

We sincerely thank the editor and the reviewers for their thoughtful and constructive feedback, which has helped us improve the quality and clarity of our work. We greatly appreciate the time and effort invested in reviewing our manuscript and providing detailed comments and suggestions. Below, we address each comment point by point and describe the corresponding changes made to the manuscript starting with Additional Editor Comments. We are confident that these revisions have significantly strengthened the work and hope they meet the reviewer’s expectations.

Additional Editor Comments

The second round of reviews is now complete. My opinion is that the manuscript must undergo a major revision before considering for publication in PLOS ONE.

What emerges is that the manuscript lacks a unitary style. In addition, the abstract and the introduction should (at least) mention all the algorithms used in the experiments. In this respect, the abstract and the introduction fail in presenting the research.

We revised the style and writing according to the reviewers’ suggestions. We added the GNNs to the abstract and introduction (see Editor request).

Reviewers' comments:

Reviewer's Responses to Questions

Comments to the Author

1. If the authors have adequately addressed your comments raised in a previous round of review and you feel that this manuscript is now acceptable for publication, you may indicate that here to bypass the “Comments to the Author” section, enter your conflict of interest statement in the “Confidential to Editor” section, and submit your "Accept" recommendation.

Reviewer #3: (No Response)

Reviewer #4: All comments have been addressed

2. Is the manuscript technically sound, and do the data support the conclusions?

Reviewer #3: Yes

Reviewer #4: No

3. Has the statistical analysis been performed appropriately and rigorously?

Reviewer #3: Yes

Reviewer #4: Yes

4. Have the authors made all data underlying the findings in their manuscript fully available?

Reviewer #3: Yes

Reviewer #4: Yes

5. Is the manuscript presented in an intelligible fashion and written in standard English?

Reviewer #3: Yes

Reviewer #4: Yes

6. Review Comments to the Author

Reviewer #3

The manuscript presents an original study that applies Graph Neural Networks (GNNs) to predict sepsis using Complete Blood Count (CBC) data. This is an interesting and promising approach, but the paper has several areas that require improvement. While the second half of the paper addresses many initial issues, there are gaps in clarity, the experimental setup, and interpretability that need to be tackled before the manuscript can be considered for publication.

#R3_1: Originality

The application of GNNs to patient-centric graphs for sepsis detection is novel and contributes to ongoing research into machine learning in healthcare. However, the work overlaps with previous studies like Chen et al. (Eur Rev Med Pharmacol Sci 2021; 25 (14): 4693-4701, DOI: 10.26355/eurrev_202107_26380), which used GNNs for pediatric sepsis. A more explicit comparison with this and similar works, along with a justification of the broader patient population, would strengthen the claim of originality.

Answer #R3_1: We included the reference and highlighted that they classified pediatric sepsis instead of sepsis in adults, only used similarity graphs and incorporated multiple laboratory tests instead of using only data from the clinical routine like complete blood count data (“[GNNs] already showed promising results for predicting pediatric sepsis based on several groups of laboratory tests (e.g., medical history and serological tests) using similarity graphs [17].”see #R3_1).

#R3_2: Clinical Relevance

The selection of CBC parameters is reasonable, but the relevance of each feature to sepsis prediction should be better explained. For readers unfamiliar with clinical data, it would be helpful to briefly introduce why these specific markers are key indicators of sepsis and to support this with appropriate references.

Answer #R3_2: We added a supplementary note to explain the functions of individual blood parameters and their potential relevance for sepsis. A text reference for the supplementary note was added in the manuscript (“The functions and the potential relevance of the individual blood parameters for sepsis is discussed in the supplement (Supplementary Note 1)”, see #R3_2).

#R3_3: Ethical Considerations

The manuscript states that ethical approval was not applicable due to the anonymization of the data. However, further elaboration is necessary regarding how the privacy of patient data was ensured. Data privacy is particularly sensitive in healthcare applications, and a brief explanation of how the dataset was anonymized would help meet ethical transparency standards.

Answer #R3_3: We added ethical approvals from the original study of Steinbach et al. (“The Ethics Committee at the Leipzig University Faculty of Medicine approved the initial study from Steinbach et al. [26] (reference number: 214/18-ek). The study was published in accordance with the transparent Reporting of a multivariable prediction model for Individual Prognosis or Diagnosis (TRIPOD) statement. This study is only re-evaluating the dataset from Steinbach et al. [26] by evaluating GNNs and incorporating time-series information“, see #R3_3).

Evaluation of Paper Sections

#R3_4:• Abstract

The abstract is concise but could be clearer. For instance, it mentions AUROC as the primary evaluation metric but doesn’t justify why this metric was chosen over others like the F1 score, which is more commonly use. A brief justification for focusing on AUROC would be helpful. Additionally, more details about the graph structure (why it was used over simpler models) and how GNNs handle time-series data would improve the clarity.

Answer #R3_4: We added respective details in the abstract (“Methods: In this study, we evaluated the performance and time consumption of several GNNs (e.g., Graph Attention Networks) on similarity graphs compared to simpler, state-of-the-art machine learning algorithms (e.g., XGBoost) on the classification of sepsis from blood count data as well as the importance and slope of each feature for the final classification. Additionally, we connected complete blood count samples of the same patient based on their measured time (patient-centric graphs) to incorporate time series information in the GNNs. As our main evaluation metric we used the Area Under Receiver Operating Curve (AUROC) to have a threshold independent metric that can handle class imbalance.

Results and Conclusion: Standard GNNs on evaluated similarity-graphs achieved an Area Under Receiver Operating Curve (AUROC)[...]”, see #R3_4).

#R3_5:• Introduction

The introduction is somewhat lacking in depth, particularly concerning the clinical background of sepsis and the rationale for using GNNs. A more detailed explanation of why GNNs are suited for sepsis prediction, compared to simpler methods, would benefit the reader. The discussion of DeepWalk and Node2Vec compares them to GNNs, implying they are machine learning algorithms. These methods are more accurately described as graph embedding techniques used for feature representation, not prediction. This section would benefit from clarification and more technical precision. The introduction should also be expanded to give a clearer description of how the graph structure was defined. The definition of nodes and edges is somewhat imprecise and could confuse readers unfamiliar with graph theory terminology. For example, terms like "vertices" and "links" are interchangeable with "nodes" and "edges," but their use should be consistent.

Answer #R3_5: We corrected our definition of a graph and clarified that DeepWalk and node2vec are embedding techniques. Then, we added more details regarding the clinical background of sepsis. Finally, we added more details on how we defined the graph structures and the rationale behind using GNNs (“A graph G is a non-empty finite set of elements called nodes V(G) and finite set E(G) of distinct unordered pairs of distinct elements of V(G) called edges [6]. [...]GNNs have the advantage that they can utilize attached features and parallelize computations on modern hardware (e.g., GPUs) in contrast to using embedding techniques like DeepWalk [15] or Node2Vec [16]. […] Sepsis is a life-threatening organ dysfunction caused by a dysregulated immune response to an infection [28]. The inflammatory response is driven through the release of cytokines from neutrophil granulocytes and macrophages. Blood parameters like white blood cells, red blood cells, platelets, hemoglobin and mean corpuscular volume might serve as easily available indicators for sepsis [26] (Supplementary Note 1). ”, see #R3_5).

#R3_6: • Methods

The methods section provides a detailed explanation of the experimental setup but lacks justification for some key decisions. For instance, why were tree-based algorithms chosen as baselines? Further, the reasoning behind certain hyperparameter choices, such as the number of epochs and learning rate, is unclear. More justification for selecting GraphSAGE as the representative GNN is necessary: were other GNN architectures considered, and if so, why was GraphSAGE chosen as the final model?

Answer #R3_6: We used tree-based and non-tree based (e.g., logistic regression and a neural network) as benchmarks to get a comprehensive overview of the performance evaluation across several algorithms. We also added justifications for chosen hyperparameters and why we chose GraphSAGE as final GNN model for the interpretability (“As benchmarks, we evaluated the performance (AUROC, F1- Macro Score, and MCC) on tree-based and non-tree-based algorithms (Fig. 1 C) to get a comprehensive performance evaluation across several algorithms. […] The high epoch number in combination with a low learning rate should prevent under-fitting. Early-stopping on a separate validation set was used to prevent over-fitting. […] GraphSAGE was chosen as the final model since it achieved a reliable classification performance on the homogeneous and heterogeneous similarity graphs.”, see #R3_6).

#R3_7: Results

The results section is one of the paper’s strengths, providing a comprehensive evaluation of the models and a comparison between different machine learning approaches. However, the choice of AUROC as the primary metric remains problematic. The authors do mention F1 later in the paper but should provide more justification for emphasizing AUROC initially. The partial dependence plots in the results are well-executed, but the explanation of how the direction of individual features (e.g., an increase in white blood cell count) affects predictions could be clearer. While the paper mentions this limitation, providing more interpretability around feature impact is crucial, particularly in clinical settings where decisions depend on understanding how individual factors contribute to risk.

Answer #R3_7: We added justifications why chose AUROC as primary evaluation metric (“In the following, we will mainly focus on AUROC as the primary evaluation metric to have a threshold-independent evaluation metric that can also incorporate the high class imbalance. By assessing model performance across different thresholds, AUROC enables clinicians to fine-tune the sensitivity and specificity of sepsis detection according to their needs, minimizing both the risk of missing septic patients and the potential harm from overdiagnosis, such as unnecessary antibiotic treatments.”, see #R3_7). Additionally, we provided clearer explanations for the partial dependence plots (“Specifically, that means older people with higher white blood cell counts, red blood cell counts and increased corpuscular volume have higher sepsis probabilities according to the trained models. White blood cells had a minimum of around 4 – 8 Gpt/l (Giga- particles per liter) for most algorithms (Fig. 3 D) that indicates a physiological white blood cell count at this range. In contrast to the positive gradient, we observed a negative gradient for platelets for all algorithms (Fig. 3 G), indicating decreased sepsis risk for rising values according to the models. Specifically, this means that patients with lower platelets (thrombocytopenia) have a higher sepsis probability according to the trained models. Tree-based algorithms (Decision Tree, RUSBoost, Random Forest, XGBoost) do not depend (gradient near zero) on the features hemoglobin (Fig. 3 C) and red blood cells (Fig 3 E) in contrast to non-tree-based algorithms (Logistic Regression, Neural Network, Homogeneous GNN, Heterogeneous GNN). The latter ones showed an increased sepsis probability for low hemoglobin levels (anemia) (Fig. 3 C). Finally, the feature “sex” is nearly irrelevant for all algorithms (i.e., low gradient), i.e., the sepsis probability does not significantly depend on sex (Fig. 3 B). ”, see #R3_7). In terms of interpretability, we are actively developing a novel graph learning framework aimed at significantly improving interpretability, which remains a focus of our ongoing work.

#R_3_8: • Discussion

The discussion section does a good job of addressing the limitations of the study, including bias in the dataset and computational challenges. However, the paper would benefit from a more detailed analysis of the computational complexity of GNNs, which are known to be resource-intensive. Were any optimizations considered, such as using more efficient GNN architectures or limiting the number of layers to reduce computational overhead?

Answer #R3_8: We added a description for the general time complexity of GNNs (“In general, the computational complexity of a single convolution layer of a GNN is O(VFF’ + EF’), where V represents the number of nodes in the graph, E the number of edges in the graph, F the number of input features and F’ the hidden dimension or output features for the convolution [8,43]. However, the computational complexity differs across different architectures, depending on factors like the number of attention heads (e.g., in GAT and GATv2), and used multi-layer perceptrons (e.g., in GIN). ”see #R3_8). The number of layers was already quite low with only two layers. One layer was evaluated in initial experiments but showed a significantly lower classification performance. In terms of reducing computational overhead, as stated above we are currently working on a new graph learning framework that reduces computational complexity compared to Graph Neural Networks (GNNs) while significantly enhancing interpretability of results.

• Data and Code Availability

The transparency in making the code and data publicly available on GitHub a

---

## [Decision Letter · Decision Letter 2]

Edges are all you need: Potential of Medical Time Series Analysis on Complete Blood Count Data with Graph Neural Networks.

PONE-D-24-06777R2

Dear Dr. Walke,

We’re pleased to inform you that your manuscript has been judged scientifically suitable for publication and will be formally accepted for publication once it meets all outstanding technical requirements.

Kind regards,

Qiang He

Academic Editor

PLOS ONE

Additional Editor Comments (optional):

we acknowledge that the paper now addresses the criticism raised by reviewers.

Gladly, we recommend acceptance

Reviewers' comments:

Reviewer's Responses to Questions

**Comments to the Author**

Reviewer #4: All comments have been addressed

Reviewer #5: All comments have been addressed

2. Is the manuscript technically sound, and do the data support the conclusions?

Reviewer #4: Yes

Reviewer #5: Yes

3. Has the statistical analysis been performed appropriately and rigorously?

Reviewer #4: Yes

Reviewer #5: Yes

4. Have the authors made all data underlying the findings in their manuscript fully available?

Reviewer #4: Yes

Reviewer #5: Yes

5. Is the manuscript presented in an intelligible fashion and written in standard English?

Reviewer #4: Yes

Reviewer #5: Yes

Reviewer #4: The authors revised the paper according to reviewers' suggestions. Therefore I recommended the publication in correct form.

Reviewer #5: The authors have effectively addressed my initial concerns. The study’s originality is clarified by distinguishing its focus on routine CBC data in adults and providing a meaningful comparison to prior work. Clinical relevance is improved with supplementary material on the role of CBC parameters, and ethical considerations are addressed through clarification of anonymization and ethical approval. The abstract and introduction now provide stronger justification for using AUROC and GNNs for time-series data. Methodological justifications, including the choice of GraphSAGE and baseline algorithms, enhance the rigor of the experimental setup. Results are more interpretable with improved explanations of partial dependence plots, and the discussion addresses computational complexity and planned optimizations.

**Do you want your identity to be public for this peer review?** For information about this choice, including consent withdrawal, please see our Privacy Policy

Reviewer #4: No

Reviewer #5: No

---

## [Editor Report · Acceptance letter]

PONE-D-24-06777R2

PLOS ONE

Dear Dr. Walke,

I'm pleased to inform you that your manuscript has been deemed suitable for publication in PLOS ONE. Congratulations! Your manuscript is now being handed over to our production team.

Kind regards,

on behalf of

Dr. Qiang He

Academic Editor

PLOS ONE